# Deriving fine-scale models of human mobility from aggregated origin-destination flow data

**Constanze Ciavarella**[1]*, **Neil M. Ferguson**[1,2]

**1** MRC Centre for Global Infectious Disease Analysis, School of Public Health, Imperial College London,
**2** The Abdul Latif Jameel Institute for Disease and Emergency Analytics (J-IDEA), School of Public Health,
Imperial College London

* ciavarella.constanze@gmail.com

Deriving fine-scale models of human mobility from
aggregated origin-destination flow data. PLoS
Comput Biol 17(2): e1008588. https://doi.org/

UNITED STATES

**Data Availability Statement:** The full analysis code
is available at https://github.com/ConniCia/RP01_
spatial_fit_code. Input data is freely available from
third-party providers as detailed in the following.
Kenyan flow data are available from the published

## Abstract

The spatial dynamics of epidemics are fundamentally affected by patterns of human mobility. Mobile phone call detail records (CDRs) are a rich source of mobility data, and allow semi-mechanistic models of movement to be parameterised even for resource-poor settings. While the gravity model typically reproduces human movement reasonably well at the administrative level spatial scale, past studies suggest that parameter estimates vary with the level of spatial discretisation at which models are fitted. Given that privacy concerns usually preclude public release of very fine-scale movement data, such variation would be problematic for individual-based simulations of epidemic spread parametrised at a fine spatial scale. We therefore present new methods to fit fine-scale mathematical mobility models (here we implement variants of the gravity and radiation models) to spatially aggregated movement data and investigate how model parameter estimates vary with spatial resolution. We use gridded population data at 1km resolution to derive population counts at different spatial scales (down to $\sim$ 5km grids) and implement mobility models at each scale. Parameters are estimated from administrative-level flow data between overnight locations in Kenya and Namibia derived from CDRs: where the model spatial resolution exceeds that of the mobility data, we compare the flow data between a particular origin and destination with the sum of all model flows between cells that lie within those particular origin and destination administrative units. Clear evidence of over-dispersion supports the use of negative binomial instead of Poisson likelihood for count data with high values. Radiation models use fewer parameters than the gravity model and better predict trips between overnight locations for both considered countries. Results show that estimates for some parameters change between countries and with spatial resolution and highlight how imperfect flow data and spatial population distribution can influence model fit.

## Author summary

The growing use of large-scale individual-based models calls for reliable modelling of human population movement at ever finer scales. Mobility models have at times been fit to fine-scale movement data, such as travel questionnaires and GPS data. However, the

article "Quantifying the Impact of Human Mobility on Malaria" by Wesolowski et al. DOI:10.1126/science.1223467 (https://science.sciencemag.org/content/suppl/2012/10/10/338.6104.267.DC1). Namibian flow data are available from the published article "Identifying Malaria Transmission Foci for Elimination Using Human Mobility Data" by Ruktanonchai et al. DOI:10.1371/journal.pcbi.1004846 (https://doi.org/10.1371/journal.pcbi.1004846.s002). Population distribution data are freely available from LandScan (https://landscan.ornl.gov). Spatial data on administrative units are freely available from GADM (https://gadm.org).

**Funding:** CC and NMF acknowledge funding from the MRC Centre for Global Infectious Disease Analysis (reference MR/R015600/1), jointly funded by the UK Medical Research Council (MRC) and the UK Foreign, Commonwealth & Development Office (FCDO), under the MRC/FCDO Concordat agreement and is also part of the EDCTP2 programme supported by the European Union. CC was supported by the Wellcome Trust (https://wellcome.ac.uk/) [203851/Z/16/Z]. NMF acknowledges funding by Community Jameel. The funders had no role in study design, data collection and analysis, decision to publish, or preparation of the manuscript.

**Competing interests:** The authors have declared that no competing interests exist.

restricted size of such datasets make them suboptimal for parametrising large-scale simulations. Larger datasets, such as census commuting data or mobile phone data, pose a different problem in that such datasets are usually made available at a much coarser spatial resolution than required for individual-based simulations. Here we present a straightforward, if computationally intensive, method to obtain fine-scale movement estimates from coarse-scale movement data. We trial the method on movement data from Kenya and Namibia and implement two of the most common mathematical mobility models, the gravity and the radiation models. Our findings confirm previous research that the parameter estimates for the mobility models differ across spatial scales and countries. We also investigate how population spatial distribution and the characteristics of the flow datasets influence parameter estimates.

## Introduction

Over the past few years, individual-based models have been widely adopted across multiple scientific disciplines [1–8]. Such models simulate population processes providing a set of rules that determine the behaviour of individuals in the population depending on their state and on the interactions they have with other individuals in the model. The recent growing availability of computational power and large datasets allows for detailed and accurate modelling of synthetic populations.

Data on human mobility is particularly important to accurately capture the spatial range of interactions such as disease transmission. Typical data sources include census data on migration and commuting, satellite imagery, surveys, airline and other long-distance travel ticketing records [9–11]. Each of these sources provides travel data at different spatio-temporal resolutions and has specific sampling biases; nonetheless, they inform mobility models in a wide variety of fields [12], including epidemiological modelling [13–16].

Most recently, the rapid uptake of mobile phones even in low-income countries has provided a new, very rich source of data on human mobility. Call Detail Records (CDRs) store the time and routing mobile phone tower of each incoming or outgoing communication (call or text message). Depending on the timespan of the data, it is possible to infer movement patterns in a number of different ways [17]. CDRs are subject to specific biases [18, 19]: (1) ownership is heterogeneous across regions, age groups and socio-economic classes, (2) usage may vary widely between individuals. Despite this, mobile phone ownership is widespread even in resource-poor settings such as sub-Saharan Africa [20].

Due to privacy concerns, freely available mobile phone data is spatially aggregated and fine-scale data is accessible only through special agreements [15, 17, 21]. Aggregated mobility data is not immediately suitable to be employed in individual-based models that represent space at a much finer scale. In this work, we present new methods to fit models of human mobility to spatially aggregated data on movement patterns. We will analyse a gravity model and several variants of the radiation model, allowing the spatial resolution of the models to be higher than that of the data itself. Past work on UK and US commuting data [13] suggests that the parameter estimates for gravity models change with the scale at which space is discretised; we examine whether similar effects can be resolved with the two CDR-based datasets from Kenya and Namibia we analyse here.

The mobility models we fit have a wide range of potential applications; while our motivation is to inform the modelling of the transmission of diseases such as malaria, similar models

are widely used in multiple other areas, such as the planning of service distribution, both for commercial and healthcare purposes, and in the managing and maintenance of transport infrastructures.

## Results

Specifying mobility models at finer spatial scales resulted in a considerable increase in computational complexity; halving the spacing of the spatial grid caused an almost 4-fold increase in the number of cells with non-zero population (Table 1).

Analysis of the population distribution showed that the Kenyan population tends to aggregate in larger centres, whereas the Namibian population is more dispersed (S1 Fig). This characteristic is accentuated at finer spatial resolutions. At 5km scale for Namibia, 52% of the cells have a population of 10 or less, accounting for 2.4% of the Namibian population. In contrast, only 4.2% of cells at the 5km scale in Kenya have 10 or fewer individuals, accounting for only 0.01% of the total population.

We implemented one version of the gravity model, GM, and four variants of the the radiation model, in order of complexity RM1, RM2, RM3, RM4 (see Materials and methods for details).

For Kenya the grid scales perform better than the administrative unit scale; however, for Namibia the opposite is true (Fig 1A). If we only look at the gridded scales, model fit generally increases as grid scale decreases; the exception to this being the fit of the radiation models RM1-RM4 to the Kenyan dataset and model RM1 to the Namibian dataset (S1 Text). As a result, models fit the Namibian dataset best at the administrative unit scale, while the Kenyan dataset is best fit by the gravity model at the 5km scale, and by the radiation models at the 20km scale.

Radiation models RM1-RM4 clearly outperform the gravity model in terms of log-likelihood in both settings and across all scales (S1 Text), except for grid-scale models RM1 and RM2 for Namibia. The difference between the log-likelihoods of the gravity model, GM, and the best-fitting radiation model, RM4, is especially large for the Namibian dataset.

We sampled 100 combinations from the posterior distributions of the parameters of models GM and RM4 across all spatial scales and models. We computed the expected origin-destination flows for each sample and computed the average and 95% credible interval (CrI) for each. The simulated flows compare reasonably well to the input data (Fig 2). While both models seem to predict large flows relatively well (Fig 2, yellow and green pixels), they overestimate low and medium counts (Fig 2, blue and dark blue pixels). For Kenya, model RM4 predicts large and medium flows well, but overestimates small flows. The gravity model, GM, reproduces very large flows somewhat better, but underestimates medium and low flows. For Namibia, model GM overestimates very large and very small flows, whereas the radiation model RM4 does better with large flows, but still somewhat overestimates small ones and underestimates medium ones. Observed flow counts fall rarely within the 95% CrI of the

**Table 1. Number of cells by spatial scale and country.** We excluded cells with no population. The administrative units are reported at level 1 for Kenya and level 2 for Namibia.

| Scale | Kenya | Namibia |
|---|---:|---:|
| Administrative unit | 68 | 90 |
| 20 km | 1 793 | 2 472 |
| 10 km | 6 792 | 8 348 |
| 5 km | 26 001 | 23 126 |

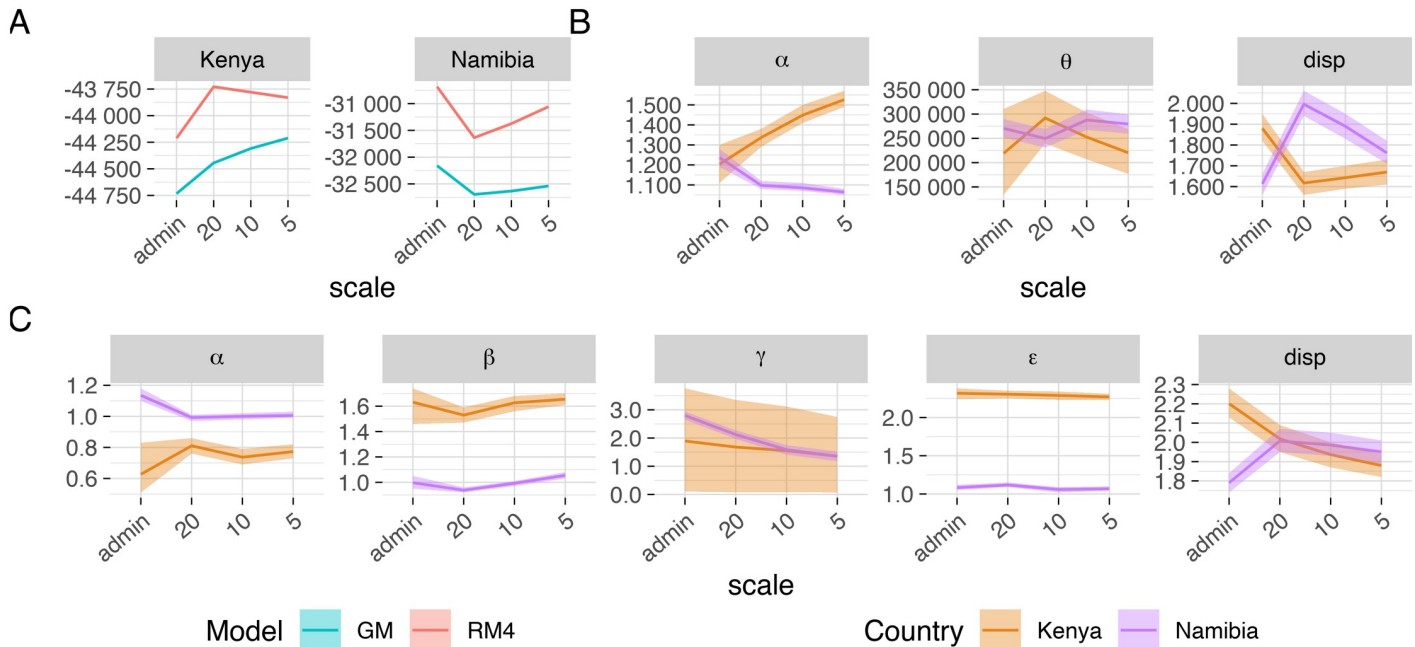

**Fig 1. Mean value and 95% credible interval (CrI) of log-likelihood and fitted parameters.** Mean value and 95% credible interval (CrI) of log-likelihood and fitted parameters of the gravity model, GM, and the best-fitting radiation model, RM4, for Kenya and Namibia at varying spatial scales across all 4 MCMC chains. Parameters shown are the increment on the origin population, $\theta$, the power on the origin population, $\alpha$, the power on the destination population, $\beta$, the distance scale, $\gamma$, the spatial kernel power, $\varepsilon$, and the over-dispersion parameter of the negative binomial distribution, *disp*. The exact values of the mean and 95% credible interval for all parameter values, including the proportionality constant $\kappa$, are reported in the supplementary information (S1 Text). (A) log-likelihood of GM (blue) and RM4 (red) for Kenya and Namibia, (B) RM4 parameters for Kenya (orange) and Namibia (purple), (C) GM parameters for Kenya (orange) and Namibia (purple).

simulated flows for each origin-destination administrative unit pair (S1 Table), but they nevertheless give a reasonable approximation of the overall movement patterns.

Comparing the observed distribution of cumulative trip frequency by distance with that from models GM and RM4 at the best-fitting scales (Fig 3A), we see that the gravity model overestimates especially short and mid distance trips, whereas the radiation model underestimates mid distance and to a lesser extent also short distance trips.

The proportion of trips per distance follows a power law with exponent 1.91 for Kenya and 0.84 for Namibia (Fig 3B, S2 Table). The value for Kenya is broadly in line with measurements from other countries, while the exponent for Namibia is decidedly lower [22–26]. This indicates that distance is a stronger deterrent to travel in Kenya than in Namibia. Both models, RM4 and GM, reproduce the patterns and power law fit well. The power law pattern can also be recognised if we look at trip frequency by distance for the distribution of trips starting from a single origin, in this case the locations where the most (Fig 3C, S2 Table) respectively the fewest (Fig 3D, S2 Table) trips start.

## Discussion

In this work, we explored the ability of a range of popular mathematical models of human mobility to reproduce observed patterns of movement in Kenya and Namibia. A key aim was to develop methods to fit such mobility models at a higher spatial resolution than the observed mobility data. Models are applied directly when we fit models at the same spatial scale as the mobility data, i.e. at administrative unit scale. In order to fit mobility models at a finer scale, we define models on gridded representations of geographic space at 20km, 10km and 5km

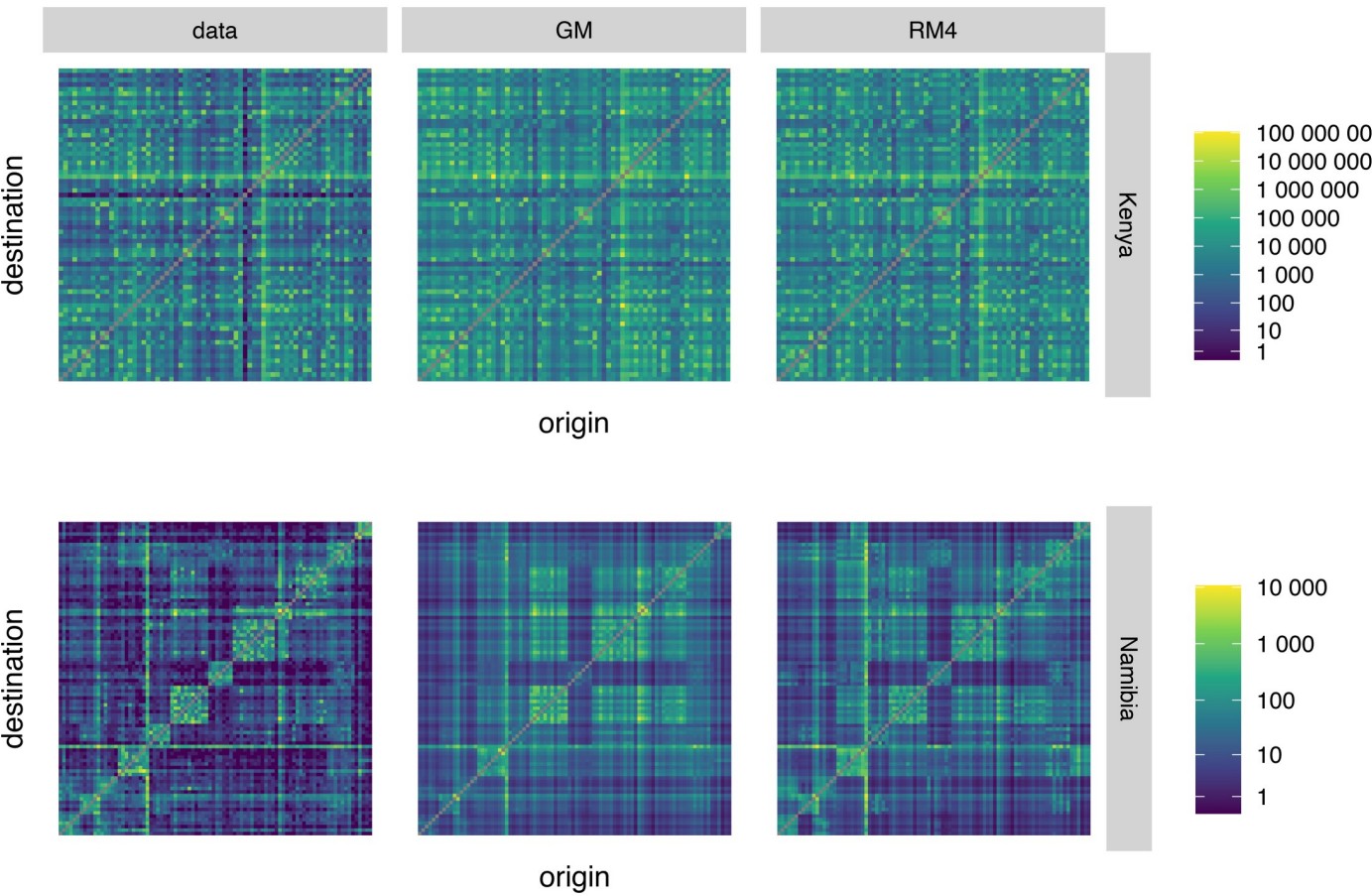

**Fig 2. Simulated origin-destination flows.** Origin locations are plotted along the y-axis, destination locations against the x-axis. Each pixel of the plot represents the average estimated flow across 100 parameter combination samples from the posterior distributions between the respective origin and destination administrative units. The Kenyan dataset did not report on within-unit trips, and radiation models do not predict within-unit trips. For this reason, the pixels on the diagonal, representing within-unit flows, have been removed for all models and settings. Top row: plots relative to Kenya; bottom row: plots relative to Namibia. Left column: input flow data; central column: best-fitting scale of the gravity model GM (5km scale for Kenya, administrative unit level for Namibia); right column: best-fitting scale of radiation model RM4 (20km for Kenya and administrative unit level for Namibia).

scale and we sum over all expected flows between grid squares that are associated with a particular origin and destination administrative unit pair.

We implemented a gravity model and several versions of the radiation model adapted for trips between overnight locations [13, 14, 16], since we are mostly interested in applications to malaria control and elimination. We equipped the gravity model with a power law distance kernel designed to fit long-distance trips [13]. The main text describes four different radiation models, more models and their results are described in the supplementary information S1 Text. The simplest radiation models RM1 and RM2 are based on the original parameter-free formulation given by [27]. Models RM3 and RM4 are based on adaptations put forward by [28] to better fit trips between overnight locations; RM3 and RM4 introduce an additional parameter to RM1 and RM2, respectively. Classic radiation models require knowledge of the proportion of travellers in the population. This proportion is then assumed to be the same for each origin location. Neither of the datasets we used provided this population-wide datum; therefore, we estimated this quantity from the data. We implemented the classic approach assuming a constant proportion of travellers across all origin locations (models RM1 and RM3), and more sophisticated hypotheses (models RM2 and RM4) assuming e.g. a power law

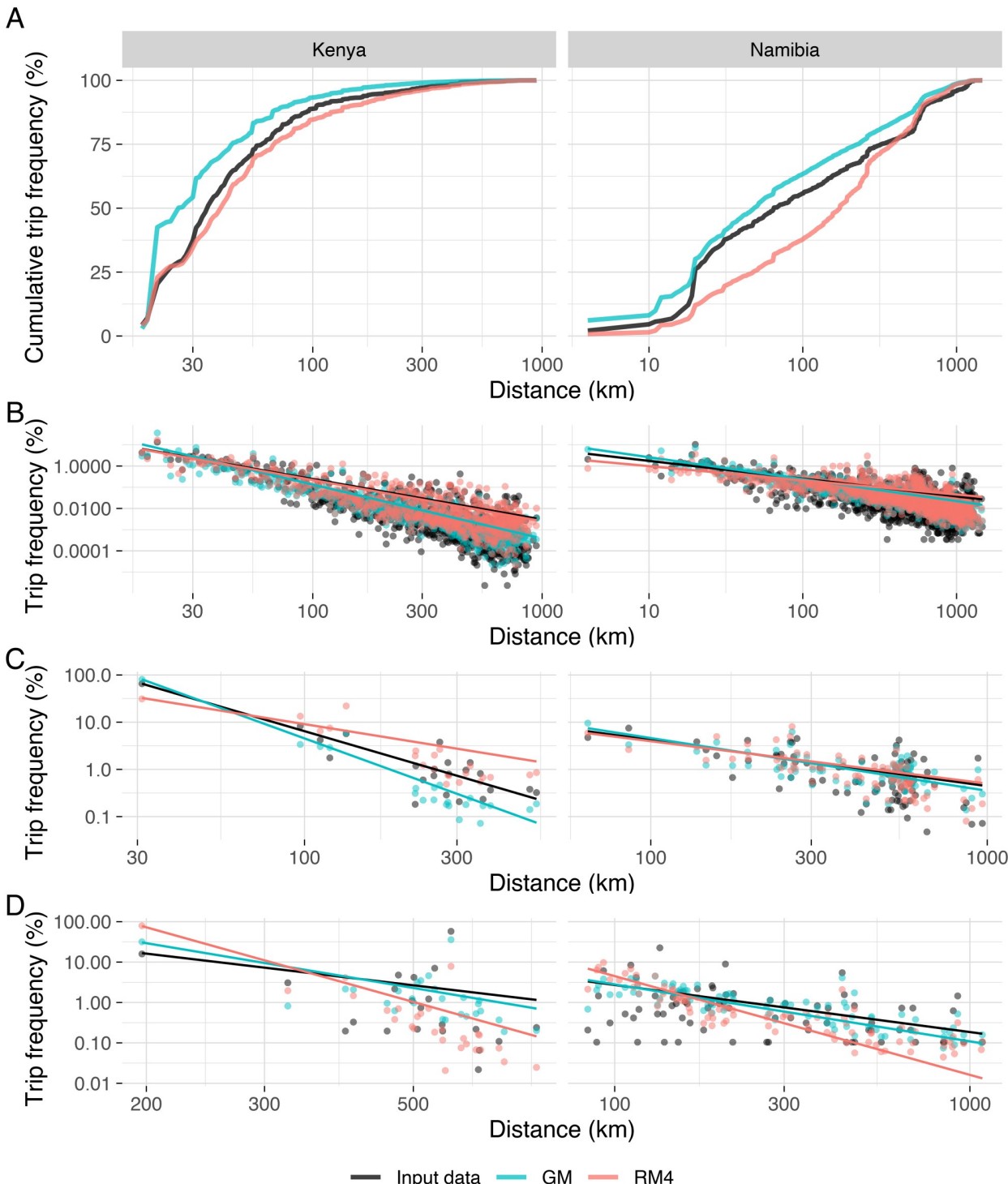

**Fig 3. Trip frequency by distance.** Empirical data (black) and simulated data using the gravity model, GM, (blue) and the best-fitting radiation model, RM4 (red). For Kenya, we use the 5km scale for GM and the 20km scale for RM4. For Namibia, we use the administrative unit level for GM and RM4. Left column panels relate to Kenya, right column panels to Namibia. (A) Cumulative trip frequency by distance across all origin-destination pairs. (B) Trip frequency by distance across all origin-destination pairs (points) and power law fits (lines). (C) Trip frequency by distance restricted to trips originating in the administrative unit where the most trips start (Nairobi Province (now Nairobi County) for Kenya, the union of Windhoek rural and Windhoek west constituencies of the Khomas region for Namibia) and power law fits (lines). (D) Trip frequency by distance restricted to trips originating in the administrative unit where the fewest trips start (Moyale District (now part of Marsabit County) for Kenya, Uuvudhiya Constituency of the Oshana region for Namibia) and power law fits (lines).

dependence on the origin population. Overall, the gravity model has five parameters, whereas the radiation models have between one and three parameters. In addition to these, we estimate the negative binomial dispersion parameter *disp* for all models.

[14] have also fitted model RM1 to the same Kenyan data. However, the authors constrained the proportionality constant, $\kappa$, to be between 0 and 1. In this formulation of the model, $\kappa$ represents the proportion of the population that travels outside of their administrative unit of residence. Constraining this parameter is correct if we fit data that contains trips commenced on a specific day, e.g. commuting data recounting the typical work location visited in the previous week, where each individual has made up to one trip. The Kenyan travel data has been collected across a year and hence contains many more trips than the population size of Kenya. Simply allowing $\kappa$ to be greater than 1 allows for a strikingly better fit, and eliminates much of the model's shortcomings as described by the authors.

MCMC chains converge well and consistently when started from different initial conditions. Radiation models generally fit better than the gravity model (Fig 1, S1 Text). The difference in log-likelihood between the gravity model, GM, and the best-fitting radiation model, RM4, is most pronounced for Namibia, suggesting that the gravity model is less adequate to fit datasets with a sparse population. The more sophisticated radiation models, RM3 and RM4, perform better than the gravity model (S1 Text). The simpler radiation models, RM1 and RM2, fit better than the gravity model for the Kenyan dataset at all scales and for Namibia at the administrative unit scale. The ability of the radiation models to fit data from heterogeneous as well as sparse, homogeneous population better than the gravity model, highlights their greater flexibility compared with the gravity model. This finding is particularly notable given that even the most sophisticated radiation model we tested has only three parameters compared to the gravity model's five.

If we look at gridded scales alone, the log-likelihood generally improves as the grid is made finer with the exception of radiation model RM4 for Kenya (Fig 1). For Kenya, grid scales fit better than the administrative unit scale, whereas for Namibia the opposite holds true. [13] similarly notice that gravity models fit better on the coarsest spatial scale they implemented and speculate this to be due to the level of heterogeneity in population distribution at finer scales. Since our finer spatial scales are gridded, we hypothesise the difference between the administrative unit scale and the gridded scales to be due to the way the grids split up the population. Moreover, for Namibia another reason could be that we are fitting the number of days spent at each destination administrative unit rather than the number of trips, thereby likely exaggerating the number of long-distance trips in the data.

See supporting information S1 Text for a short guide on how to interpret the parameter values for the negative binomial distribution and the human mobility models. Estimates for the dispersion parameter, *disp*, of the negative binomial distribution are consistently greater than 0 for all datasets and spatial scales (Fig 1, S1 Text). Parameter values are estimated in the range 1.61–2.20, indicating considerable levels of over-dispersion in the datasets. Application of a Poisson likelihood to such data (i.e. counts with high values) is therefore unlikely to be appropriate.

Parameter values obtained fitting the gravity model, GM, differ substantially between the two assessed countries (Fig 1C, S1 Text). For Kenya, the symmetrised model cannot distinguish between the power on the origin and destination populations, $\alpha$ and $\beta$. However, there is a clear difference in the estimates for these two values. If we assume $\alpha > \beta$ as in Fig 1C, then the estimates of power on the origin population, $\alpha$, are consistent across spatial scales and are an indicator of the high travel propensity of individuals from large urban centres. The value of the power of the destination population, $\beta$, on the other hand is well below one, suggesting that high population locations attract a proportionally lower number of travallers. Should we

instead assume $\alpha < \beta$, then the estimates for $\alpha$ would be below one, indicating that individuals from lower population density areas have a higher propensity to travel than individuals from high population density areas. In that case, estimated values for $\beta$ would be well above one and be an indicator of the attractiveness of high population density areas as journey destinations. We chose to impose $\alpha > \beta$, but it has to be kept in mind that it is not possible to determine which estimates truely pertain to $\alpha$ and which to $\beta$. The underlying mobility data indicates that there is a consisten flow of travellers to and from the two major cities of Kenya, Nairobi (population of 3.4 million) and Mombasa (population of 1.2 million); however, it is not possible to distinguish between travelling residents and visiting travellers. Estimates for the distance scale, $\gamma$, are fairly low and question the necessity of the parameter itself, suggesting a pure power law kernel be used instead. The spatial kernel power, $\varepsilon$, is consistently estimated at approximately 2.3, which broadly agrees with estimates obtained by the authors of the source paper for the Kenya dataset in subsequent studies [14]. These parameter estimates are lower than the typical estimates of 3.9-4.5 obtained using UK or US commuting data [13]. At first sight, this seems to indicate that distance is a bigger deterrent to travel in the US and the UK than in Kenya, even though the former countries have a vaster transport network. However, this incongruence might be explained by the fact that the estimates for the US and the UK were obtained by fitting only to work-related trip counts, which might be biasing the data towards journeys with a shorter average distance.

Gravity model parameter estimates for the source population power, $\alpha$, obtained for Namibia are slightly above 1 for the administrative unit scale and close to 1 for the grid scales, suggesting individual's origin population density has little or no influence on their propensity to travel. Estimates for $\beta$ are consistently around 1, indicating that attractiveness of destinations varies linearly with the size of the destination population. Namibia is very sparsely populated (less than 3 inhabitants per $km^2$ and a total population of 2.4m) and the capital, Windhoek, only has a population of 325 000 inhabitants (in 2011). Hence, there are few locations that are able to attract travellers. The distance scale, $\gamma$, is estimated at values too low to play any role in the gravity model. Hence, also for the Namibian dataset the distance kernel resembles a pure power law, with the exception of within-cell flows. Estimates for the spatial kernel power scale, $\varepsilon$, are consistently close to 1. A power close to 1 yields uncommonly large numbers of long-distance journeys. This could be due to the fact that we are fitting to the number of days spent at each administrative unit rather than the number of trips made.

Parameter values obtained with the best-fitting radiation model, RM4, also vary between Kenya and Namibia (Fig 1B, S1 Text). Estimates for the increment on the origin population, $\theta$, are relatively similar for both countries, at between 221 000 and 288 000. For Namibia, these estimates seem to be consistent across scales, whereas for Kenya estimates seem to decrease with decreasing grid spacing. Conversely, parameter estimates for the power on the origin population, $\alpha$, show differing trends. Estimates using the Kenyan dataset show an increase from 1.21 to 1.53 as spatial scale is made finer, whereas estimates for Namibia indicate a decrease from 1.24 to 1.06. This suggests that for Kenya, individuals from higher population density areas have a higher propensity to travel, but that this trend is much less pronounced for the lower population density setting of Namibia.

Parameter estimates for the two models vary substantially between Kenya and Namibia. [13] and [16] have found, similarly, that parameter estimates for the gravity and radiation model were not comparable between different countries. Estimates for the dispersion parameter, $disp$, of the negative binomial distribution are clear evidence of over-dispersion supporting the use of negative binomial instead of Poisson likelihood for count data with high values. The spatial kernel power $\varepsilon$ is estimated at higher values for Kenya than for Namibia, suggesting that distance is a bigger deterrent to travel in Kenya than in Namibia. But this seems unlikely, as in both

countries just under 17% of roads are paved [29, 30]. Moreover, Euclidean distance, like we used, has been shown to yield better fits than travel time or road distance in Kenya [14]. We thus hypothesise that this uncommonly large number of long-distance trips produced by the gravity model for Namibia are again due counting too many long-distance trips in the data.

Comparing heatmaps drawn using observed origin-population flows with heatmaps obtained from modelled flows show how the models are able to pick up most of the patterns present in the data (Fig 2). However, it should be noted that all the models we tested have a poor record of including the observed flow data in the credible intervals of the modelled data (S1 Table). Models are hence good at estimating the relative importance of travel routes, but not the actual flow counts.

When comparing the modelled flows with observed flows, we found a good overlap between the two models (S2 Fig). Both models tend to overestimate low trip counts. [14] confirm this finding for the gravity model on Kenya, but they reach the opposite conclusion for the radiation model whose proportionality parameter, as we discussed, has been needlessly constrained. For very high trip counts on the top right of the panels, we see that the gravity model slightly underestimates high trip counts. [27] also observed this phenomenon, albeit more pronounced, on data from the US; however, [14] produced the best fit for the gravity model on high trip counts.

The distance profiles produced by the models highlight their major shortcomings (Fig 3). The gravity model is prone to overestimating the number of short-distance trips, while the radiation model underestimates these. The opposite holds true for long-distance trips, whose number the radiation model overestimates, and the gravity model underestimates. [27] draw similar conclusions about the gravity model missing most long-distance and some medium-distance trips. [13] note how the gravity model fails to pick up rare long-distance trips that might play an important role in infectious disease transmission. We cannot reproduce results seen in [16] who state that radiation models fair better at shorter and gravity models at longer distances.

The assumption underpinning this analysis, is that large scale human travel is driven by the same mechanisms as small scale travel. In other words, we are assuming that metrics on population distribution are sufficient to describe travel propensity across all spatial scales. A lack of very fine scale travel data make it impossible to verify this assumption empirically.

Radiation models do not estimate within-administrative unit trips, and the Kenyan dataset did not provide data on within-unit travel. Hence, we were only able to fit within-administrative unit trips to Namibian data via the gravity model. At the grid scales this meant that, for most of the implemented models, we did not estimate trips between grid cells belonging to the same administrative unit.

The growing use of large-scale individual-based models calls for reliable modelling of human population movement at ever finer scales. Mechanistic mobility models aim to estimate flows between discrete locations and thereby condensate potentially vast amounts of travel data in a compact mathematical formula and parameter estimates. This provides a huge advantage as mathematical formulae are more easily tractable and consume less computational power when employed in a model of infectious disease transmission. Moreover, if fit to sufficiently large datasets, mathematical mobility models would be able to avoid overfitting and thereby reduce the noise in the travel patterns. However, these models have to be fitted to the empirical data they are trying to approximate. Imperfect data can hugely distort predictions made by mathematical mobility models and hence need to be analysed more closely. In this research, we have shown how mobility model parameter estimates are influenced by (1) the scale at which space is discretised (Fig 1), (2) the population distribution and geographic conformity of a country (S2 Text), (3) the method by which the movement data has been collected and

processed (S2 Text), (4) the positioning of the spatial grid (S2 Text), and (5) whether we have partial or complete knowledge of the movement data for the geography of interest (S2 Text).

The dependence of estimates for the origin population power on spatial scale (Fig 1) suggests that the finest resolution considered here, 5km, may still be too coarse to give reliable parameter estimates for use in fine-scale, i.e. <1km scale, simulation models of infectious disease transmission. Thus further research to explore how estimates change as one further refines the spatial scale down to the 1km level is a priority. However, model fitting at the 1km scale would encounter prohibitive computational costs using the methods we described in this chapter; hence, further methodological innovation is needed. A random sampling algorithm of origin-destination pairs could be employed in order to reduce computational complexity. We have shown how this could reach satisfying results in the supporting information S2 Text, but would need to evaluate this method at finer spatial scales.

The dependence of parameter estimates on the geography and population density of a country can cause important variations in the log-likelihood and parameter estimates of the gravity and radiation models (S2 Text). Moreover, Kenyan and Namibian estimates remain incompatible even following efforts to improve the similarity between their spatial population distribution. This means that parameter estimates computed for a specific country cannot be transferred to another country, thus effectively preventing the implementation of mobility models when these cannot be fit to observed mobility data.

The quality and means of extraction of movement data clearly influence parameter estimates and log-likelihood (S2 Text). This makes data quality and consistency across datasets from different countries a prime concern. Symmetricity of travel flows has the greatest effect on log-likelihood values and parameter estimates. Since prime sources of movement data are hard to access, mobility models are often forced to use secondary data that has been extracted using often very different methods to define trips and trip origins [17, 21]. Further research into the relationship between the quality of the observed mobility data and parameter estimates is a priority.

Parameter estimates and goodness of fit are mostly unaffected by changes in the positioning of the spatial grid (S2 Text); however, in some cases they have a noticeable impact. Dividing spatial scale into regular squares might be convenient for computational purposes, but is a highly unusual geography on which to observe human movement. The consequences of spatial grids splitting densely inhabited metropoles or sparsely populated rural lands have to be further investigated.

Travel data that is freely available to researchers might at times be patchy, i.e. only detailing trip counts for a subset of possible travel routes (S2 Text). Sampling 15 random locations from the Kenyan and Namibian dataset achieves satisfying log-likelihood estimates, but the estimates of the powers on the origin and destination popopulations vary somewhat from the default estimates. Increasing the sample size to 30 or even 50 locations considerably improved log-likelihood and parameter estimates. A few of the parameter estimates had larger credible intervals; however, their means were similar to those yielded by the default simulations.

Analyses in this project used freely available CDR-derived data that is aggregated to the administrative unit scale and lacks temporal data altogether. Even though this provides a valuable insight into travel at national level, it lacks fine-scale spatial and temporal detail. Hard-to-access raw CDR data is a much richer source of information. CDR data records time and location (at the granularity of the mobile phone tower) of a large number of anonymised subscribers, and can thus be used to extract information on the spatial range, duration and periodicity of journeys. Combining CDR data with questionnaires on travel and mobile phone usage potentially allows to associate CDR-based travel patterns with specific demographic groups for

targeted public health interventions [28]. Access to fine-scale CDR data would hence greatly increase our understanding of the spatial and temporal patterns of human movement. At the same time it will be a priority to develop more sophisticated mobility models that account for timing and periodicity of travel, and heterogeneity in patterns of travel between individuals and between population groups.

Past decades have seen a considerable increase in human travel, even in low-income countries [19]. As a consequence, pandemics spread wider and faster [31], and promising elimination efforts of widespread endemic diseases are hindered by continuous reimportation from disease foci [17, 21]. Accurate modelling of human movement is thus a priority in epidemic modelling. Indeed, including movement data directly into models of disease spread is not possible where data is patchy or is provided at a coarser scale than the epidemic transmission model.

## Materials and methods

### Mobility data

We analysed two CDR-based datasets, one for Kenya [17] and one for Namibia [21], that are published as population flows between administrative units. Both datasets only record trips that resulted in overnight stays, since the authors were primarily interested in malaria transmission, which takes place mostly at night through mosquito bites [32].

The Kenyan dataset counts the journeys of almost 15 million subscribers (roughly 32% of the population at the time of data collection) between 69 districts (administrative unit 1 level, Fig 4 left) from June 2008 to June 2009. The data does not inform on within-district travel and lacks directionality: the subscriber's administrative unit of residence is not known nor inferred, making it impossible to distinguish between outbound and return journeys. This is confirmed by inspecting the origin-destination matrix, which is close to symmetric (S3 Fig).

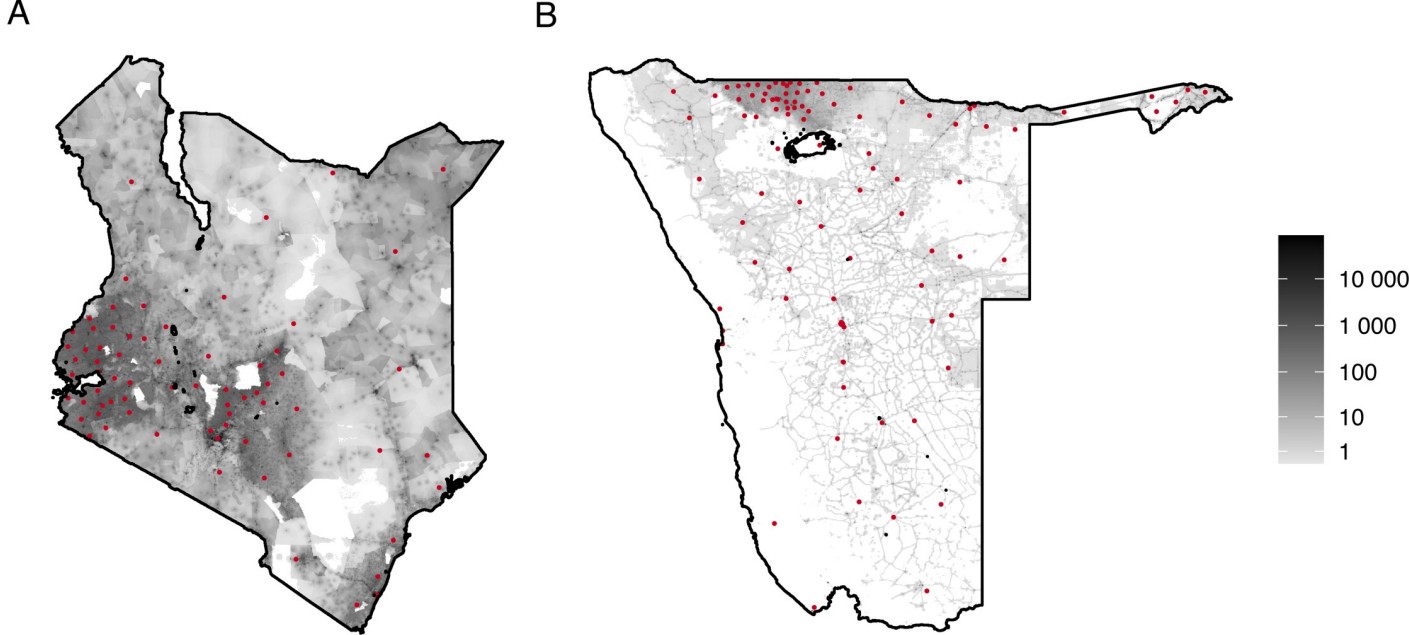

**Fig 4. Population density per $\sim 1\text{km}^2$.** Population counts are plotted on a grey log-scale by spatially aggregating $\sim 100 \times 100$ m level data obtained from WorldPop [33, 34]. Red dots represent population-weighted centroids for the administrative unit level.

The Namibian data contains journeys between 107 constituencies (administrative unit 2 level, Fig 4 right) of 1.19 million subscribers (about 54% of the population at the time of data collection) from October 2010 to September 2011. This data is provided as the proportion of nights residents of a specific constituency spend in each constituency (including their constituency of residence). To obtain an integer count of journeys, we multiply these values by the number of residents in the respective constituency. As a result, we count each night a resident of a specific constituency spends in a different administrative unit as an outbound journey, thereby exaggerating the number of journeys made. Administrative units that are not present in the mobility dataset or have zero ambient population, have not been included in the analysis.

## Population data

We used population density data from LandScan [35] and administrative boundary information from GADM [36] to compute the ambient population for the administrative units present in the mobility datasets (Fig 4), and for each square cell resulting from grids of 20km, 10km and 5km spacing over each country. For all administrative units and cells, we computed the population-weighted centroids and the distance (in metres) between all centroids. We associated each cell with the administrative unit of most of its area. A number of administrative units for Namibia were not associated with any grid cell at one or multiple grid scales as they were too small. For consistency, we wanted to fit cells associated with the same list of administrative units across all spatial scales. Therefore, we merged administrative units that failed to be associated with any cell in at least one of the grid scales with their closest neighbouring administrative unit across all grid scales.

## Mathematical models of human mobility

Gravity and radiation models are among the most commonly used mathematical models of human movement. While both models have their origin in physics, they are conceptually distinct. Gravity models borrow from classical mechanics and adapt Newton's law of gravitation to describe population flows as depending on the population of origin and destination locations, as well as some function of the distance between them [16, 37]. Radiation models are influenced by the emission-absorption process of electromagnetic radiation: atoms (origin locations) emit particles (individuals) in all directions; these can be absorbed by other surrounding atoms (destination locations) depending on their energy level (the population within the same distance from the origin location) [16, 27]. Thus, the radiation model does not explicitly employ distance in its equations.

We implemented a gravity model and a range of radiation models adapted for trips between overnight locations [13, 14, 16, 27]. The main text focuses on results for the gravity model and four radiation models; a description and results for further radiation model variants can be found in the supplementary information together with a short guide on how to interpret the model parameters introduced in the following (S1 Text).

**Gravity model.** The gravity model we use here describes the expected population flow between all possible origin and destination locations. The flow, $GM_{ij}$, from location $i$ with population $m_i$ to location $j$ with population $m_j$ at a distance $d_{ij}$ is assumed to be proportional to

$$\frac{m_i^\alpha m_j^\beta}{f(d_{ij})}. \tag{1}$$

For our implementation of the gravity model, we pick a spatial kernel function, $f(d_{ij})$, designed to penalise long-distance trips by reducing the probability at which they occur [27]. An offset power law,

$$f(d_{ij}) = 1 + \left(\frac{d_{ij}}{10^\gamma}\right)^\varepsilon, \tag{2}$$

represents a common choice [38] and has been shown to fit human mobility flows well at long distances [16, 22].

Note that the gravity model (1) is a directional model: it describes outflows of individuals resident in $i$ to $j$. This is appropriate for Namibia where we define the following model

$$GM_{ij}^{(N)} := 10^\kappa \, \frac{m_i^\alpha m_j^\beta}{1 + \left(\frac{d_{ij}}{10^\gamma}\right)^\varepsilon}. \tag{3}$$

Here, $\kappa$, $\alpha$, $\beta$, $\gamma$ and $\varepsilon$ are parameters to be estimated.

Since the Kenyan dataset does not distinguish between outbound and return journeys, the 'origins' in that dataset do not necessarily correspond to individuals' residential administrative units. To allow for this, we sum journeys from $i$ to $j$ and journeys from $j$ to $i$ in the mobility dataset and define a symmetrised gravity model

$$GM_{ij}^{(K)} := 10^{\kappa - \gamma\varepsilon} \, \frac{(m_i^\alpha m_j^\beta + m_j^\alpha m_i^\beta)}{1 + \left(\frac{d_{ij}}{10^\gamma}\right)^\varepsilon}. \tag{4}$$

Given that the Kenyan dataset does not inform on within-admin travel, we choose the factor $10^{\kappa - \gamma\varepsilon}$ to enhance convergence of parameter $\kappa$ (S4 Fig). Indeed, note that we can rewrite the non-symmetrised gravity model (3) as

$$GM_{ij}^{(N)} = \frac{10^{\kappa + \gamma\varepsilon} \, m_i^\alpha m_j^\beta}{10^{\gamma\varepsilon} + d_{ij}^\varepsilon}. \tag{5}$$

Now, whenever $i = j$, the distance vanishes and the expression simplifies to

$$GM_{ii}^{(N)} = 10^\kappa \cdot 2 \cdot m_i^{\alpha + \beta}. \tag{6}$$

Thus, the within-patch flows provide information to infer $\kappa$. However, for the Kenyan dataset, we never compute $GM_{ii}^{(K)}$ because we have no corresponding within administrative unit travel data to fit it to; thus, for values of $\gamma$ and $\varepsilon$ for which $10^{\gamma\varepsilon} \ll d_{ij}^\varepsilon$ (as is the case for our Kenya estimates), only the expression $\kappa + \gamma\varepsilon$ can be inferred, as is seen in the right hand panels of S4 Fig.

**Radiation model.** The radiation model proposed by Simini et al. in 2012 [27] does not explicitly depend on distance but rather the radial distribution of population about the origin location. In its original formulation the flow from location $i$ with population $m_i$ to location $j$ with population $m_j$ is defined as

$$T_i \frac{m_i \, m_j}{(m_i + r_{ij} + m_j) \, (m_i + r_{ij})}, \tag{7}$$

where $T_i$ is the number of travellers resident in location $i$ and $r_{ij}$ is the population contained in a ring centred at location $i$ and of radius equal to the distance between $i$ and $j$ (N.B. $r_{ij}$ excludes $m_i$ and $m_j$).

We also examined a variation of the model that downweighted local journeys by artificially increasing the origin population by an amount $\theta$ [16]

$$T_i \frac{(m_i + \theta)\, m_j}{(m_i + \theta + r_{ij} + m_j)\, (m_i + \theta + r_{ij})}. \tag{8}$$

For the datasets we analyse here, the number of travellers, $T_i$, per administrative unit is unknown and has to be estimated from the data thereby introducing further parameters that have to be estimated. Combining models (7) and (8) with different estimates for $T_i$, we evaluated the following models (in order of complexity):

$$RM1_{ij}^{(N)} := \sum_{ij} \kappa\, m_i\, \frac{m_i\, m_j}{(m_i + r_{ij} + m_j)\, (m_i + r_{ij})}, \tag{9}$$

$$RM2_{ij}^{(N)} := \sum_{ij} 10^\kappa\, m_i^\alpha\, \frac{m_i\, m_j}{(m_i + r_{ij} + m_j)\, (m_i + r_{ij})}, \tag{10}$$

$$RM3_{ij}^{(N)} := \sum_{ij} 10^\kappa\, m_i\, \frac{(m_i + \theta)\, m_j}{(m_i + \theta + r_{ij} + m_j)\, (m_i + \theta + r_{ij})}, \tag{11}$$

$$RM4_{ij}^{(N)} := \sum_{ij} 10^\kappa\, m_i^\alpha\, \frac{(m_i + \theta)\, m_j}{(m_i + \theta + r_{ij} + m_j)\, (m_i + \theta + r_{ij})}. \tag{12}$$

Here, $\kappa$, $\alpha$ and $\theta$ are parameters to be estimated.

Alternative hypotheses for the radiation model and the estimate for the number of travellers have been investigated, but fitted less well (see Materials and methods, S1 Text). The above equations are valid for the directional flows of the Namibian dataset. The implementation of the Kenyan variant is again symmetrised to match the symmetry seen in the data.

## Fitting fine-scale mobility models to aggregated mobility data

The models described above are used directly when we fit at the same spatial resolution as the mobility data, i.e. at administrative unit level. To fit the mobility models at a finer scale than the mobility data, we sum over all flows between cells that are associated with a particular origin and destination administrative unit pair. Taking $\{c_i\}$ as the set of fine scale cells associated with administrative unit $i$, the gravity model (3) for Namibia becomes

$$GM_{ij}^{(N)} := \sum_{c_i, c_j} \frac{10^\kappa\, m_{c_i}^\alpha m_{c_j}^\beta}{1 + \left(\frac{d_{c_i,c_j}}{10^\gamma}\right)^\varepsilon}, \tag{13}$$

where $m_{c_i}$ and $m_{c_j}$ denote the populations of the fine scale cells $c_i$ and $c_j$ respectively, and $d_{c_i,c_j}$ is the distance between fine-scale cells $c_i$ and $c_j$. Analogous modifications apply to all other models.

We fit the parameters of the gravity and radiation models to the observed flows between administrative units using Latin Hypercube Sampling and Markov Chain Monte Carlo (MCMC) methods. For each model, we performed Latin Hypercube Sampling to find suitable candidates to start 4 MCMC chains with. Each MCMC chain ran for 1.3 million iterations. We computed results only on the converged portion of each chain.

We assume a negative binomial likelihood function with dispersion parameter *disp* to describe the count data of flows between administrative units

$$\mathcal{L}(Model \mid Data) =$$

$$= \prod_{ij} \frac{\Gamma(Data_{ij} + 1/disp)}{Data_{ij}! \, \Gamma(1/disp)} \left( \frac{Model_{ij}}{Model_{ij} + 1/disp} \right)^{Data_{ij}} \left( \frac{1/disp}{Model_{ij} + 1/disp} \right)^{1/disp}, \quad (14)$$

where *i* and *j* are administrative units.

The choice of the negative binomial (as compared with a Poisson distribution) allows for possible over-dispersion in the mobility data. Presence of over-dispersion in the data is indicated by $disp > 0$. Conversely, if $disp = 0$, the negative binomial coincides with a Poisson distribution.

Specifying mobility models at a 5km scale resulted in prohibitive computational costs, forcing us to exclude cells with up to 10 individuals from the model. The computational gain derived from removing low population cells was considerable for Namibia, but only limited for Kenya (S1 Fig).

## Supporting information

**S1 Text. Expanded methods and results.** More in-depth description of the models discussed in the main text (GM and RM1-RM4) together with additional radiation models implemented as part of this project. Additional tables for parameter interpretation, goodness of fit of all implemented models, and parameter estimates for all implemented models.
(PDF)

**S2 Text. Sensitivity analyses.** Description and parameter estimates for a range of sensitivity analyses. Additional graphs for sensitivity analyses: simulated origin-destination flows and trip frequency by distance.
(PDF)

**S1 Table. Proportion of origin-destination flow counts that fall within the 95% credible intervals (CrIs) of the estimated flows.**
(PDF)

**S2 Table. Estimated parameter values of the power law fit in Fig 3 of the main text.**
(PDF)

**S1 Fig. Distribution of cell population across spatial scales.** Top row: cumulative frequency of cells by their population; bottom row: cumulative proportion of population contained in cells with a log population size smaller or equal than the values reported on the x-axis. Columns refer to different spatial scales: administrative unit, 20km, 10km and 5km scales. Data for Kenya is in orange, data for Namibia in purple.
(PDF)

**S2 Fig. Empirical vs. modelled flow counts.** We plotted the best-fitting scale of the gravity model, GM, (5km scale for Kenya, administrative unit level for Namibia) and the best-fitting scale of radiation model, RM4, (20km for Kenya and administrative unit level for Namibia). Modelled flow counts are computed as the mean across the flows resulting from 100 parameter combinations sampled from the posterior distributions of the models. The Kenyan dataset did not report on within-unit trips, and radiation models do not predict within-unit trips. (A) Kenya, (B) Namibia. Note that since the mobility models for Kenya are symmetric, we plot

each point twice and therefore they appear to be darker in panel A than in panel B.
(PDF)

**S3 Fig. Symmetry of the origin-destination matrices for Kenya (left) and Namibia (right).**
Each pixel represent an origin-destination administrative unit pair. Origin locations are plotted along the x-axis, destination locations against the y-axis. Let $i$ and $j$ be an origin and a destination location, and let $f_{ij}$ be the flow from $i$ to $j$ as reported in the empirical data. For each $i$ and $j$ we compute $F_{ij}$: $= (f_{ij} - f_{ji})/max(f_{ij}, f_{ji})$ and plot it. This yields a value between -100 and 100: red and orange pixels (values $> 25$) indicate that the journeys from $i$ to $j$ where considerably more numerous that the journeys from $j$ to $i$, dark blue and bluish pixels (values $< -25$) indicate that the journeys from $j$ to $i$ where considerably more numerous that the journeys from $i$ to $j$, yellow pixels (-25 $<$ values $< 25$) indicate that the journeys form $i$ to $j$ where about as numerous as the journeys from $j$ to $i$ demonstrating thus that the flow between the two administrative units is symmetric. For Kenya the absolute value of values $F_{ij}$ is 11.04 (95%CrI 0.00-42.86), whereas for Namibia the average lies at 45.59 (95%CrI 0.00-100.00).
(PDF)

**S4 Fig. Trade-off between parameters $\kappa$ and $\gamma$ for the gravity model, GM, for Kenya.** Left column: default model fitted for Kenya (as described in the main text); right column: we implemented the same model with the exception that the model scaling factor was $10^{\kappa}$. The 10km and 5km grid scales were not fit for the model in the right column.
(PDF)

## Acknowledgments

We thank Dr. Wes Hinsley for providing technical support.

## Author Contributions

**Conceptualization:** Neil M. Ferguson.

**Data curation:** Constanze Ciavarella.

**Formal analysis:** Constanze Ciavarella, Neil M. Ferguson.

**Funding acquisition:** Neil M. Ferguson.

**Investigation:** Constanze Ciavarella.

**Methodology:** Constanze Ciavarella, Neil M. Ferguson.

**Project administration:** Neil M. Ferguson.

**Resources:** Neil M. Ferguson.

**Software:** Constanze Ciavarella.

**Supervision:** Neil M. Ferguson.

**Validation:** Constanze Ciavarella.

**Visualization:** Constanze Ciavarella.

**Writing – original draft:** Constanze Ciavarella.

**Writing – review & editing:** Neil M. Ferguson.

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
