## [Decision Letter · Decision Letter 0]

21 Oct 2019

Dear Dr Ciavarella,

Thank you very much for submitting your manuscript 'Deriving fine-scale models of human mobility from aggregated origin-destination flow data' for review by PLOS Computational Biology. Your manuscript has been fully evaluated by the PLOS Computational Biology editorial team and in this case also by independent peer reviewers. The reviewers appreciated the attention to an important problem, but raised some substantial concerns about the manuscript as it currently stands. While your manuscript cannot be accepted in its present form, we are willing to consider a revised version in which the issues raised by the reviewers have been adequately addressed. We cannot, of course, promise publication at that time.

Sincerely,

Cecile Viboud

Associate Editor

PLOS Computational Biology

Rob De Boer

Deputy Editor

PLOS Computational Biology

[LINK]

Reviewer's Responses to Questions

**Comments to the Authors:**

Reviewer #1: In “Deriving fine-scale models of human mobility from aggregated origin-destination flow data”, the authors use aggregated mobility data to develop a finer spatial scale model of mobility. The authors build off of the two most commonly applied spatial interaction models (gravity model and radiation model) and use those frameworks to estimate travel at a finer spatial scale. The authors use population and distance data to estimate these travel patterns (down to 1 km). Using previously published mobility data calculated using mobile phone calling records, the authors first fit spatial interaction models to the origin-destination trip counts. The authors the use these origin-destination trip counts to assume that the sum of travel on finer spatial scales is equal to the aggregate. By assuming the finer scale data is equal to the sum of the aggregate, they then fit the same spatial interaction models to these data. Overall, the analysis presented is adequate but would be greatly strengthened by 1) elaborating on the implications and interpretation of the models fit, 2) providing quantitative results of the implications of these finer scale models, and 3) demonstrating the implications for a biological purpose. The primary concern for the methods is the omission (not the fault of the authors) of within administrative unit travel and the implications on the parameters estimated.

Major Comments:

a) The authors do not analyze any travel within a spatial unit. Although the authors state that the within administrative unit data is unavailable (based on the published work where the authors obtained these data), this omission (which is not the fault of the authors) may be biasing estimates from the models. In particular, the role of distance and the estimated parameter (since very short distances – or distances of length 0) may be impacted. On the larger spatial scale of the data, this omission may be less relevant since travel is likely to occur over larger distances (given the administrative unit geography). However, this assumption (that within spatial unit travel can be omitted from the analysis), may be less valid for the finer spatial units analyzed. For example, on the 1km grid scale, very local travel is highly likely since these represent very close and small spatial units. Including data on these scales would likely down weight the distance parameter. However, travel patterns on these scales are not included since they were not included in the originally published data set. Hence the authors cannot state that they are estimating travel between these fine spatial units when actually they are estimating travel between fine spatial units assuming that travel only occurs between large spatial units. This fact underpins the entire analysis, yet the authors do not carefully and clearly address these points in the analysis.

b) The assumption in the analysis is that the same factors such as population size and distance are driving the heterogeneity of travel for finer spatial units. For example, if X trips occur between two districts (or aggregated administrative unit), then the assumption is that the number of trips (that will sum to X) that occur from the grid cells comprising those districts are also driven by population size and distance factors. Since trip data on finer spatial units is unavailable to directly fit parameter estimates, it is unclear if these factors or other factors are the most relevant data sets.

c) The authors should provide additional support and analyses of the model fits. Currently, enough detail is not provided. In particular, a more thorough comparison and investigation of the estimated model fits versus the data should be provided since it forms the primary result.

d) In general, the conclusions do not appear to be the same for Kenya and Namibia. They find opposite results in Kenya and Namibia but do not discuss the implications of these findings further.

e) And in particular, the finer spatial scale models do not perform better in Namibia. The authors should provide additional analyses and interpretation of this finding. Further, in the discussion the authors discuss the sparseness of the population distribution on finer scales. It should be possible to analyze this hypothesis further to provide quantitative support. It is unfair to state in the abstract and results that the finer-spatial models outperformed the aggregate models.

f) The authors do not provide any interpretation of the model parameters or the differences on various spatial scales. Given that this is the one of the primary results of the manuscript, additional analyses and interpretation should be provided.

g) The authors do not provide any implications for biological applications. Although the overall motivation is to study infectious disease dynamics, they do not provide any results that evaluate the impact of different models on disease dynamics. For example, the radiation model may outperform the gravity model in some instances, however these differences may not be relevant for disease transmission given the life history of the pathogen.

h) In the gravity model, the authors have added additional parameters (such as 10^kappa – gamma*epsilon) but do not provide additional support or interpretation for the added these parameters.

Minor Comments:

a) The authors should provide higher resolution figures since the current versions, although interpretable, are blurry to view.

b) There are a number of typos in the manuscript particularly regarding verb tense.

c) It is unclear exactly how the bottom row of figures in Figure S3 was calculated.

d) The authors should provide code to reproduce their analysis and the predicted model output.

Reviewer #2: In the proposed article, the authors fit different models (gravity and radiation models) to CDR data at different levels of spatial resolution using datasets from Kenya and Namibia. The suitability of often-used gravity models is only partially understood and appears to depend on spatial resolution and across locations. This paper provides an approach to explore these questions. I only have a few comments.

The appropriateness of the different model formulations depends on the quality of the CDR data used to fit the models. This appears particularly problematic for the Kenyan dataset, as there was no direction data available for the flows. The authors come up with an approach that sums over both directions, however, the uncertainty generated by such an approach was not clear to me. As a sensitivity analysis, the authors could aggregate the Namibian dataset to make an equivalent to the Kenyan one (ie with no direction information) and use the same approach. This would at least provide some insight as to the consistency of the results and the potential pitfalls of losing direction information.

It was unclear how the authors dealt with within-unit travel – in particular, what distance did they use in the gravity models (e.g., distance of 0, 0.5 x cell width, population weighted average based on LandScan……)? – I could imagine this could be quite important to model fit as most travel with be within the same unit.

It was difficult to assess model fit – the summary measures do not explore the fit of individual district pairs. The authors could plot the observed number of trips between each district pair with that predicted from the different models (aggregating the smaller units where necessary). That would allow readers to assess whether there are any systematic differences.

Relatedly, it would also be good to understand the robustness of the fit to held-out datasets (e.g., fitting in one part of the country and estimating in another)

**Have all data underlying the figures and results presented in the manuscript been provided?**

Reviewer #1: No: The authors do not provide their code and the model output.

Reviewer #2: No: Not provided directly but available from previous publications

PLOS authors have the option to publish the peer review history of their article (what does this mean?). If published, this will include your full peer review and any attached files.

Reviewer #1: No

Reviewer #2: No

---

## [Decision Letter · Decision Letter 1]

1 Dec 2020

Dear Ms. Ciavarella,

We are pleased to inform you that your manuscript 'Deriving fine-scale models of human mobility from aggregated origin-destination flow data' has been provisionally accepted for publication in PLOS Computational Biology.

Best regards,

Cecile Viboud

Associate Editor

PLOS Computational Biology

Rob De Boer

Deputy Editor

PLOS Computational Biology

Reviewer's Responses to Questions

**Comments to the Authors:**

Reviewer #2: The authors have done a very impressive, comprehensive reanalysis of the data, incorporating the concerns of the reviewers. The paper now does a good job of discussing the complexities of using simple models to fit data, including the limitations. I have no further comments.

**Have all data underlying the figures and results presented in the manuscript been provided?**

Reviewer #2: Yes

PLOS authors have the option to publish the peer review history of their article (what does this mean?). If published, this will include your full peer review and any attached files.

Reviewer #2: No

---

## [Editor Report · Acceptance letter]

7 Feb 2021

PCOMPBIOL-D-19-01187R1 

Deriving fine-scale models of human mobility from aggregated origin-destination flow data

Dear Dr Ciavarella,

I am pleased to inform you that your manuscript has been formally accepted for publication in PLOS Computational Biology. Your manuscript is now with our production department and you will be notified of the publication date in due course.

With kind regards,

Alice Ellingham
